# Mesenchymal Stem Cell-Derived Exosomal microRNAs in Cardiac Regeneration

**DOI:** 10.3390/cells12242815

**Published:** 2023-12-11

**Authors:** Meghana Bhaskara, Olufisayo Anjorin, Meijing Wang

**Affiliations:** Department of Surgery, Indiana University School of Medicine, Indianapolis, IN 46202, USA

**Keywords:** mesenchymal stem cells, exosomes, microRNA, cardiac regeneration, myocardial infarction

## Abstract

Mesenchymal stem cell (MSC)-based therapy is one of the most promising modalities for cardiac repair. Accumulated evidence suggests that the therapeutic value of MSCs is mainly attributable to exosomes. MSC-derived exosomes (MSC-Exos) replicate the beneficial effects of MSCs by regulating various cellular responses and signaling pathways implicated in cardiac regeneration and repair. miRNAs constitute an important fraction of exosome content and are key contributors to the biological function of MSC-Exo. MSC-Exo carrying specific miRNAs provides anti-apoptotic, anti-inflammatory, anti-fibrotic, and angiogenic effects within the infarcted heart. Studying exosomal miRNAs will provide an important insight into the molecular mechanisms of MSC-Exo in cardiac regeneration and repair. This significant information can help optimize cell-free treatment and overcome the challenges associated with MSC-Exo therapeutic application. In this review, we summarize the characteristics and the potential mechanisms of MSC-derived exosomal miRNAs in cardiac repair and regeneration.

## 1. Introduction

Mesenchymal stem cell (MSC)-based therapy is one of the most promising modalities for cardiac repair and regeneration. Accumulated evidence from clinical trials have demonstrated its safety, practicality, and potential effectiveness for treatment of ischemic heart disease [1,2,3,4]. The findings from our group and others have revealed that MSC paracrine action is the primary mechanism mediating tissue protection against ischemia [5,6,7,8,9,10,11,12]. However, poor survival, limited homing, and insufficient engraftment of implanted MSCs greatly hamper a better prognosis.

Of note, emerging studies provide evidence that MSC-derived exosomes (MSC-Exos) emulate MSC secretome-mediated cardiac repair [13]. Indeed, MSC-Exos function as regulators of MSC-mediated paracrine protection by transporting miRNAs (miRNA or miR), messenger RNAs, and proteins to target cells [13]. Among these, miRNA is a key functional form in exosomes to deliver regulatory information, thus impacting the physiology of recipient cells. miRNAs constitute an important fraction of exosome content and are key contributors to the biological function of exosomes [14,15,16]. MSC-Exos carrying specific miRNAs have been reported to mediate cardiac repair and regeneration after myocardial infarction (MI), encouraging angiogenesis and cell proliferation, while reducing apoptosis and inflammation in the ischemic heart. Therefore, the MSC-Exo serves as a potential therapeutic while circumventing the risks and disadvantages associated with cell therapies.

MI remains the leading cause of heart failure and death worldwide. Myocardial ischemia, leading to myocardial dysfunction and infarction, is responsible for the irreversible loss of functional cardiomyocytes. Therapeutic prevention of myocyte loss in infarcted myocardium potentially through cardiac regeneration is therefore valuable and a topic of well-deserved attention in research [17]. Considering the demonstrated impact of exosomal miRNAs in post-MI cardiac remodeling [18], in this review, we summarize the characteristics and the potential mechanisms of MSC-derived exosomal miRNAs in cardiac repair and regeneration.

## 2. The Characteristics of MSC-Exos

### 2.1. Characterization of MSC-Derived Extracellular Vesicles

MSC-derived extracellular vesicles (MSC-EVs) include exosomes, microvesicles, and apoptotic bodies, which package various signaling molecules for intercellular communication within a phospholipid bilayer membrane. Designations between each category are made by the size and method of secretion. The smallest of the EVs are exosomes (40–150 nm), the largest are apoptotic bodies (1000–5000 nm), and microvesicles (150–1000 nm) are of an intermediate size [19].

Among the different EVs, exosomes are produced and released using a more complex pathway. Exosomes require cargo sorting of contents into intracellular vesicles, which are stored in multivesicular endosomes until they are degraded by lysosomes or released through endosomal fusion with the cell membrane [20,21]. Microvesicles are produced by budding of the plasma membrane and contain a sampling of intracellular cytoplasm [22,23]. Apoptotic bodies are remnants of apoptotic cells after membrane blebbing and contain cytoplasmic contents for disposal, downstream signaling, material recycling, and control of immune responses by transmission of autoantigens [24].

Exosomes contain diverse cargo, including transcription factors, enzymes, growth factors, cytokines, lipids, DNA, mRNA, and non-coding RNAs [25,26]. They are involved in widely varied cellular communication functions, including enrichment of signaling pathways, immunomodulation, and protection against oxidative stress. In stimulated conditions of tissue expansion or ischemia, exosomes fortify PDGF, EGF, FGF, and NF-κB, Wnt, and TFG-β pathways [27,28]. Through indoleamine 2,3 dioxygenase, prostaglandin E2, IL-10, nitric oxide (NO), and hepatocyte growth factor (HGF), exosomes decrease inflammatory and fibrotic responses of innate and adaptive immune cells [29]. Analysis of exosomal proteins revealed antioxidant enzymes manganese superoxide dismutase (MnSOD), SOD1, SOD2, peroxiredoxin 1-6, catalase, thioredoxin, glutathione S-transferases GSTO and GSTP1. These enzymes contribute to exosomal protection against oxidative stress and cell death in the in vitro organ tissue [30]. Specific to cardiac tissue, MSC-Exo administration reduces apoptosis in cardiomyocytes undergoing stress and type 1 collagen production by fibroblasts undergoing serum starvation [28].

### 2.2. miRNAs in MSC-Exo

miRNAs are short non-coding sequences that post-transcriptionally influence gene expression. miRNAs are encoded in the genome with their own promoter sequences for transcriptional regulation. Once transcribed as pre-miRNAs, the stem-loop transcripts are processed into 20–22 nucleotide miRNAs by nuclear and cytosolic RNases, Drosha and Dicer, respectively [31]. Through sequence-dependent and sequence-independent pathways, both pre-miRNAs and processed miRNAs can be selected for inclusion in endosomes to be secreted within exosomes [32]. Though the selection mechanisms dictating inclusion of miRNAs into exosomes remain unclear, there is filtering of cytosolic components for exosomal incorporation [33]. Through this sorting process, miRNA concentration is greater in exosomes in comparison to the cytosol [34]. Indeed, it is evident that several miRNAs in MSC-Exo vary significantly in concentration from the MSCs they are derived from [13]. Levels of miR-874-3p, miR-133a-3p, miR-331-5p, miR-133c, miR-92a-3p, etc. were greatly elevated in MSC-Exo compared to their parent rat bone marrow MSCs. Our previous study also demonstrated that MSC-Exo contained enriched miR-199a-3p levels compared to the corresponding human BM-MSCs [35]. The selective qualitative and quantitative inclusion of miRNAs into exosomes strengthens the significance of the role miRNAs play in exosome functions. These higher concentrations of miRNAs may explain the greater therapeutic effect of MSC-Exo compared to MSCs in cardiac repair.

Besides selective inclusion for certain miRNAs from cytosol into exosomes, miRNA profiles in exosomes can vary by cell line, developmental state, and microenvironment of cell of origin [36,37,38]. Based on the needs of cells shaped by the stresses they undergo, production of exosomes and miRNA content varies. In vivo application shows exosome release from border cells following MI, likely reflecting altered metabolism and pathways responding to the hypoxic and apoptotic stresses posed by MI [39]. This evidence designates MSC exosomal miRNAs as valuable signaling molecules that can contribute to increased tissue resilience in response to stress. With the link between miRNA profiles in exosomes and disease state being very strong, exosome miRNA expression is under investigation for use as biomarkers or diagnostic tools for disease types and progressions [40,41,42].

miRNAs contained in exosomes have significant biological functions. After transmembrane proteins of exosomes interact with signaling receptors to be endocytosed, miRNAs are released from the exosomes into recipient cells. miRNAs bind to cytosolic mRNA transcripts through complementary pairing either perfectly to the transcript or imperfectly to 3′ UTR or ORF segments [31]. miRNAs bind the Argonaut protein of RNA-induced silencing complexes (RISCs), which leads to destruction or transcriptional silencing of the mRNA transcript bound by the miRNA [33,43]. miRNA may even act in the nucleus as a transcriptional enhancer [44]. Further non-classical roles for miRNAs in cell function include binding and altering protein function, activating Toll-like receptors, activating gene transcription, negatively regulating non-coding RNAs, acting on mitochondrial mRNA to affect mitochondrial function, and even being translated into peptides [45].

miRNA signaling has a powerful downstream influence. miRNAs have pleiotropic effects; a single human miRNA can alter the expression of over 100 genes. In fact, the expression of over 80% of human genes is thought to be influenced by miRNAs [46]. miRNAs are critical for development, as embryonic lethality is observed in the absence of functional miRNAs [47]. Additionally, they are implicated in anti-apoptotic and pro-regenerative effects of exosomes in the in vitro and in vivo studies. In these same studies, RNase treatment of exosomes to eradicate therapeutic miRNA levels abolished the observed anti-apoptotic and pro-regenerative benefits [48]. Their numerous and pivotal roles of miRNAs in cell development, differentiation, maintenance, and proliferation cannot be understated [49,50,51].

### 2.3. Exosomal miRNAs in MSCs with Different Origins

Three main sources of MSCs are bone marrow, adipose tissue, and umbilical cord cells. miRNA content in MSCs from these sources is different, with some shared elements [52,53]. Common miRNAs correlating with stem cell properties include miR-21 and miR-125b, which are abundantly expressed in bone marrow, adipose, and umbilical cord-derived MSCs [28,53]. miR-21 suppresses inflammation, inhibits cell death, and improves wound healing [54]. miR-125b reduces cardiomyocyte apoptosis by suppressing p53 and BAK1 expression in mice [55].

Through sequencing, most abundantly expressed miRNAs were determined for each of the three derivations of MSCs [15,56]. In a study of human bone marrow-derived MSC (BM-MSC) exosomes, miR-143-3p, miR-10b-5p, miR-486-5p, miR-22-3p, and miR-21-5p were the most abundantly expressed miRNAs [36]. In another analysis of BM-MSC exosomes, the top 10 miRNAs were miR-23a-3p, miR-424-5p, miR-144-3p, miR-130-3p, miR-145-5p, miR-29b-3p, miR-25-3p, miR-221-5p, miR-21-5p, and miR-125-5b [28]. Though measured miRNA levels do not align seamlessly, there is an overlap between abundantly expressed miRNAs as determined by these two studies of exosome analysis, indicating a level of replicability in BM-MSC exosome content [28,36]. In human adipose-derived MSC (AD-MSC) exosomes, miR-486-5p, miR-10a-5p, miR-10b-5p, miR-191-5p, and miR-222-3p were the most abundantly expressed miRNAs [36]. Meanwhile, human umbilical cord-derived stem cell (UC-MSC) exosomes contained high levels of miR-21, miR-23a, miR-125b, and miR-145 [37]. Different contents of miRNAs in exosomes of BM-MSCs, AD-MSCs, and UC-MSCs may suggest different therapeutic efficacies of these MSCs. Indeed, a functional analysis of exosomes highlights the following preferences in their repairing capacities: regeneration in BM-MSC-Exo, immune regulation in AD-MSC-Exo, and tissue damage repair in UC-MSC-Exo [57]. However, the benefits of each type of exosomes are not limited to a single function. Studies have shown a significant overlap between biological functions of exosomes derived from each of the three sources [58].

With different targets proposed for the various miRNAs of each origin of exosomes, it is clear that protective effects from exosomes are not due to a single pathway, but potentially due to a synergistic effect of multiple pathways being affected [59]. Some studies suggest that exosomes from all three human MSCs possess similar anti-inflammatory, angiogenesis-promoting, damage-limiting, and regenerative potential effects [58,60]. Meanwhile, other studies indicate superiority for a type of exosome based on the function evaluated, such as AD-MSC-Exo being superior to BM-MSC for angiogenesis and protection of cardiac function following MI [24,61]. Further preference for experimental use between the types of MSCs stems from ease of production. While AD-MSCs are abundant and easily derived in the human body, BM-MSCs are favored over AD-MSCs and UC-MSCs by their stability and ease of use. BM-MSCs are notable for their low microorganism contamination rate, stability through biological trials, greater number of passages, and low rates of immune activation [58].

Comparisons of existing studies are not conclusive as to whether any source of exosomes is reliably superior for a biological function. Due to their varied miRNA profiles, it is possible that specific exosome origins may be superior for specific functions, but it has also been established that other factors, such as stresses that the cells are undergoing, can alter exosome production and function. More studies on comparative composition and function under controlled conditions must be conducted before reaching a conclusion on comparative exosome properties.

## 3. Biological Functions of MSC Exosomal miRNAs in Heart Repair

### 3.1. Cell Proliferation and Anti-Apoptosis

MSC secreted products have been well established to be cardioprotective in contexts of MI and heart failure [62]. Besides paracrine protection, evidence supports that MSC-derived factors improve outcomes of cardiac lineage differentiation, angiogenesis, anti-fibrotic healing, anti-inflammation, and immune modulation in damaged cardiac tissue [63,64]. Particularly, MSC-Exo cargos have been shown to decrease apoptosis, repair the damaged cardiomyocytes, increase cardiomyocyte proliferation, and promote re-entry into the cell cycle.

A systems level analysis conducted by Ferguson et al. details the various biological effects of MSC exosomal miRNAs [28]. Considering its effect on the induction of proliferation in cardiomyocytes, miR-199a-3p is a particularly important miRNA included in the analysis. miR-199a is predicted to target 22 genes implicated in the regulation of cell death, proliferation, and cell cycle [28], which highlights the importance of miR-199a in supporting cell growth. Indeed, miR-199a-3p plays a key role in cardiomyocyte survival via activation of Akt survival pathway following simulated ischemia and reperfusion (I/R) in vitro using neonatal rat and adult mouse cardiomyocytes [65]. miR-199a-3p has also been determined to be one of few miRNAs crucial to the induction of cardiac regeneration [66]. Our study further reveals that miR-199a-3p is abundant in MSC-Exo and benefits cell survival in the heart following I/R [35].

miR-199a targets Crim1, a gene essential to the inhibition of cardiomyocyte proliferation [66], and has been found to increase cardiomyocyte proliferation by 30% [66]. Treatment with miR-199a-3p increases the percentage of EdU+ cardiomyocytes, associated with upregulated levels of cyclin genes in 7-day-old and adult rat cardiomyocytes ex vivo [66]. Similarly, MSC-Exo loaded with miR-199a significantly promotes cardiomyocyte proliferation as shown by increased EdU+ cardiomyocytes (from 18-day-old mouse hearts) in ex vivo culture and inhibits cardiomyocyte apoptosis through reduction in BAX expression [28]. Yet, p53 is another notable gene, the activity of which is altered by miRNA-199-3p. Additionally, p53 is intimately implicated in regulating apoptosis in a variety of mammalian cells including cardiomyocytes [67]. Upregulated p53 levels are related to increased cardiomyocyte apoptosis [68,69]. MSC exosomal miRNA-199a targets p53 to decrease expression of NF-κB and caspase-9, thus reducing cardiomyocyte apoptosis [28]. Additional evidence suggests that miR-199a-3p can target CABLES1 to mediate p53 suppression [70]. Of note, miR-199a is predicted to target retinoblastoma (RB1) (potentially regulating cell cycle arrest via RB1) [71], LKB1 for cell proliferation [72], and NEUROD1 for cell cycle arrest [73]. Collectively, these studies suggest that miR-199a can regulate cell proliferation, apoptosis, and cell cycle phases through multiple signaling pathways.

miR-21 is another cardio beneficial miRNA enriched in MSC-Exo. On the one hand, miR2-21-5p acts through PI3K to increase gene expression of calcium handling genes hECT, LTCC, and SERCA2, thus improving cardiac tissue contractility [74]. On the other hand, miR2-21-5p decreases BAX/BCL2 ratio, an indication of antiapoptotic activity and downregulates pro-apoptotic gene products PDCD4, PTEN, Peli1, and FasL, thereby conveying anti-apoptotic effects for cardiac repair [75,76]. miR-21a-5p in MSC-Exo cargo also acts as a major cardioprotective paracrine factor to mediate cardiac repair via control of PTEN-regulated apoptotic signaling [75].

Emerging evidence has demonstrated that MSC exosomal miRNAs improve cardiomyocyte survival via multiple signaling pathways. MSC-Exo miR-22 exerts anti-apoptotic effects through directly targeting methyl CpG binding protein 2 (Mecp2) [77]. miR-25-3p in MSC-Exo improves the viability of adult mouse cardiomyocytes that underwent oxygen-glucose deprivation in vitro and decreases apoptosis in mouse I/R hearts in vivo via reduction in FASL/PTEN pathway [78]. Overexpression of miR-30e in MSC-Exo significantly reduces infarct size, cardiac injury, and apoptosis, thus leading to improved cardiac function in the rat MI model [79]. MSC-Exo delivers miR-30e to increase cell proliferation (demonstrated by enhanced EdU+ cells) and downregulate the number of apoptotic cells in oxygen-glucose deprived H9c2 cells via the LOX1/NF-κB p65/caspase-9 axis [79]. miR-125b mediates cardio protection of MSC-Exo through modulating p53-Bnip3 signaling to reduce autophagic flux and cell death in hypoxia and serum starved neonatal mouse cardiomyocytes [80], while the exosomes from MSCs pretreated with the anti-miR-125b oligonucleotide have less benefit with larger infarct size and worsened cardiac function compared to the MSC-Exo group in a mouse MI model [80]. The expression of p53 was also downregulated by miR-221 from engineered MSCs. MSC-EV containing miR-221 significantly decreases neonatal rat ventricle cardiomyocyte apoptosis through reducing p53 and p53-upregulated PUMA [20]. Exosomal miR-221-3p from young MSCs further inhibits cardiomyocyte (H9c2 cells) apoptosis in vitro under hypoxia and serum deprivation and reduces myocardial apoptosis with improved cardiac function via the PTEN/Akt pathway in a rat MI model [81]. miR-205 is found to decrease cardiomyocyte apoptosis in neonatal mouse cardiomyocytes exposed to hypoxia [82]. Overexpressing miR-338 in MSC-Exo ameliorates cell death in H_2_O_2_-stressed H9c2 cells and improves cardiac function in a rat MI model via decreasing BAX/BCL2 [83]. Moreover, miR-1246 mitigates hypoxia-induced cell death in H9c2 cells following oxygen and glucose deprivation ex vivo and decreases cardiac tissue damage in a rat LAD (left anterior descending coronary) ligation model through targeting serine protease 23 (PRSS23) and down-activating the Snail/alpha-smooth muscle actin signaling [84]. Interestingly, MSC-Exo delivers miR-182-5p to reduce cell pyroptosis in neonatal mouse cardiomyocytes under hypoxia and in mouse I/R hearts via downregulation of gasdermin D [85].

The morbidity and mortality of cardiac damage stems from the inability of adult cardiac tissue to proliferate in order to adequately replace the damaged tissue. Rather than cell proliferation, cardiac cells mainly undergo hypertrophy to adapt to cardiac demands [86]. Repression of cell cycle activators and expression of cell cycle inhibitors contribute to maintenance of cardiac cells in the resting phase of the cell cycle. However, manipulation of the cell cycle can re-engage myocardial cells that have exited the cell cycle to enact cell division [87]. Environmental oxygen, e.g., hypoxia, has been shown to regulate postnatal cardiomyocyte cell cycle re-entry, as indicated by increased levels of p-histone H3 (a marker for G2-M progression) and Aurora B kinase at the cleavage furrow (a marker for cytokinesis) [87]. Once again, miRNAs can be used to manipulate gene expression in cardiomyocytes. miR-590-3p and miR-199a-3p effectively increase cardiomyocyte proliferation in vitro [66]. miR-294 overexpression has also been shown to stimulate cell cycle activity in cardiomyocytes through the Wee1 pathway [88]. Increased expression of Ki67, p-histone H3, and Aurora B kinase has been noted in neonatal rodent ventricular myocytes treated with miR-294 [88]. While other miRNAs also promote cardiomyocyte proliferation, miR-590-3p and miR-199a-3p conduct this without simultaneously inducing hypertrophy. Translation to in vivo models further validates the proliferative effect of miR-590-3p and miR-199a-3p. DNA synthesis and present mitotic figures confirm the role of these miRNAs to encourage re-entry into the cell cycle [66].

In vivo and in vitro models have provided evidence of the efficacy of exosomes in apoptotic resistance and cardiac healing. miRNAs have been shown to be central to the function of exosomes. An extensive review by [89] provides a collection of the cardioprotective effects of at least 12 miRNAs compiled from data published by more than 15 studies. miRNA activity in altering various cellular pathways further supports their values in cardiac repair and regeneration. Though the presented pathways through which various miRNAs act are not shared completely, some overlap signaling pathways exist, and synergistic effects may further potentiate the therapeutic effects of the miRNAs in MSC-Exo. MSC exosomal miRNAs in anti-apoptosis and cell proliferation are summarized in Table 1.

### 3.2. Angiogenic Effects on the Injured Heart

Analysis of MSC exosome properties contributing to cardioprotective effects reiterates the importance of miRNAs. Exosome miRNAs are mainstays in repairing damaged heart tissue following injury [90]. MSC-Exo contains many miRNAs predicted to regulate regenerative target mRNAs. Specifically, MSC-Exo miRNAs have been found to promote angiogenesis in the heart.

miRNA profiling and bioinformatics have revealed that the top 23 miRNAs constitute 79.1% of the exosomal miRNA content from human BM-MSCs. Among them, miR-23a-3p and miR-130a-3p target the greatest number of genes related to angiogenesis and vascular development [28]. miR-130a-3p has been shown to downregulate anti-angiogenic homeobox genes GAX and HOXA5, thus inducing angiogenic effects [91]. Only modified MSC-Exo with specific enrichment of miR-130a-3p, as opposed to unmodified MSC-Exo, leads to a capillary-like network of tubular structures in human umbilical vein endothelial cells (HUVECs) [28]. Human UC-MSC-Exo also promotes endothelial network formation by delivering miR-23a-3p to activate phosphatase and tensin homolog (PTEN)/Akt signaling [21]. Similarly, through the PTEN/Akt pathway, human MSC exosomal miR-21 is identified to facilitate angiogenesis in a rat model of MI [76]. The involvement of miR-126-enriched exosomes is found to significantly promote angiogenesis, resulting in increased density of WF-containing cells indicative of vascular endothelium along the infarct border. Administration of the exosomes also displayed a dose-dependent relationship in the promotion of angiogenesis, with miR-126 enriched exosomes inducing greater epithelial progenitor cell migration and tube-like structure formation in hypoxic conditions than was observed in either regular MSC-Exo or control treatments [92].

The entities of specific exosomal miRNAs reported to promote cardiac repair through angiogenesis vary between different studies. This is to be expected due to the vast number of miRNAs included in endosomes and the great variability in potential downstream effects for each miRNA. A current study reports that the benefits of AD-MSC-Exo in migration of microvascular endothelial cells and angiogenesis are abolished with administration of an miRNA-205 inhibitor in a murine MI model [82]. In addition, human UC-MSC exosomal miR-1246 has been noted to increase angiogenesis in HUEVCs by targeting serine protease 23 and reducing activation of the Snail/alpha-smooth muscle actin pathway [84]. miR-210 abundantly detected in mouse BM-MSC-Exo is shown to downregulate Efna3, thus improving tubulogenesis in vitro and enhancing neoangiogenesis with more capillaries in peri-infarct regions in post-MI mouse hearts [93]. Intramyocardial injection of MSC-Exo loaded with miR-132 significantly increases neovascularization in the peri-infarct zone and preserves heart function in a murine MI model [94]. MSC exosomal miR-30b has been found to promote HUVEC tube-like structure formation using the loss and gain function approach [95]. Finally, miR-223-5p is identified as an effective candidate in TSA-pretreated MSC-Exo to mediate cardio protection through enhanced angiogenesis and inhibited CCR2 activation to reduce monocyte infiltration [96]. These are a few of the likely numerous miRNAs included in MSC-Exo with cardiac reparative properties through angiogenesis. The abundance of miRNAs shown to benefit cardiac repair is a good omen for therapeutics, as loading specific miRNAs in MSC-Exo can be investigated to yield optimal combinations to activate synergistic mechanisms and pathways. MSC exosomal miRNAs in angiogenesis are outlined in Table 2.

### 3.3. Anti-Fibrosis and Anti-Inflammation

Fibrosis is the excessive formation of fibrous connective tissue, often as a result of chronic inflammation. Following MI, fibrosis contributes to adverse remodeling of the heart. Excessive fibrosis can impair the structure and function of the heart. Studies [28,97,98] have indicated that MSC-Exo is able to ameliorate cardiac fibrosis and limit adverse ventricular remodeling after MI by modulating the activity of fibroblasts and downregulating pro-fibrotic signals to reduce the production of extracellular matrix proteins, thus preventing the formation of excessive scar tissue. Inflammation following MI can increase deposition of extracellular collagen as well, leading to scar formation. Microparticle engineered inhibition of MSC pro-inflammatory secretome has been reported to reduce collagen deposition in human and murine cardiac fibroblasts [99]. Improving cardiac regeneration while limiting scar formation is important to long term cardiac repair. MSC-Exo containing different miRNAs may regulate fibrotic pathways and promote tissue repair through different pathways, including suppression of inflammation and cell death, and improvement of angiogenesis.

It is reported that miR-125a-3p is enriched in MSC-Exo and plays an important role in MSC-Exo elicited cardio protection, as shown by the increasing cardiac function and limiting adverse remodeling in mouse I/R hearts [100]. miR-125a-3p is considered to target *Tgfbr1*, *Klf13*, and *Daam1* to regulate the function of fibroblasts, macrophages, and endothelial cells, thus reducing fibroblast proliferation and activation, promoting M2 macrophage polarization to attenuate inflammation, and facilitating angiogenesis [100]. miR-126 is also found to reduce production of inflammatory cytokines, enhance VEGF, and decrease pro-fibrotic gene expression to mitigate adverse myocardial remodeling post MI in murine models [55,92].

Of note, the same exosomal miRNA could regulate different processes for cardiac repair attributable to the complex mechanism of action and different variables of the comprehensive effects. Through anti-apoptotic effects, MSC-Exo with increased miR-22 content has been shown to significantly reduce cardiac fibrosis compared to MSC-Exo with inhibition of miR-22 in a mouse MI model [77]. Also, AD-MSC exosomal miR-205 markedly decreases cardiac fibrosis via promoted angiogenesis and attenuated cardiomyocyte apoptosis [82]. miR-25-3p containing MSC-Exo is found to suppress expression of IL-1β, IL-6, and TNF-α in oxygen-glucose deprivation stressed adult mouse cardiomyocytes and in mouse I/R hearts via inhibition of EZH2 (enhancer of zest homologue 2)/SOCS3 axis [78]. Overexpressing miR-30e in MSC-Exo significantly decreases the level of fibrosis and suppresses MI-induced inflammation as shown by the increased anti-inflammatory factor CD206 in the rat infarcted myocardium in vivo [79]. MSC-EV-delivered miR-302d-3p diminishes inflammation, reduces myocardial fibrosis, and decreases apoptosis in the infarcted area through repressing the NFκB pathway-related MD2 and BCL6 levels [101]. Furthermore, miR-212-5p enriched MSC-EVs have been reported to reduce the levels of α-SMA, Collagen I, TGF-β1, and IL-1β in cardiac fibroblasts in vitro, attenuating cardiac fibrosis via inhibiting the NLRC5/VEGF/TGF-β1/SMAD axis [102].

Collectively, anti-inflammatory and anti-apoptotic effects of MSC exosomal miRNAs promote a more favorable microenvironment for tissue repair and prevent further damage in the heart during the acute phase of MI. Meanwhile, pro-angiogenic and anti-fibrotic activities of MSC exosomal miRNAs promote blood vessel formation to increase blood supply for delivery of oxygen and nutrients, thereby supporting tissue repair and regeneration in the reparative phase. Figure 1 summarizes MSC exosomal miRNAs in cardiomyocyte proliferation/survival, anti-inflammation, anti-fibrosis, and angiogenesis during cardiac repair after heart injury.

## 4. Modification of MSC Exosomal miRNAs

### 4.1. Preconditioning

MSCs can be manipulated for intentional delivery of miRNAs of interest through exosomes. As described above, physiological conditions and cell type alter the composition of exosomes secreted [27,28,103]. This process of altering cellular conditions to change cellular gene expression, and therefore exosome compositions, is termed “preconditioning” [26]. Also, as described previously, endosomal sorting mechanisms, though not well known, are responsible for selecting the cargo of exosomes stored within the multivesicular body during exosome synthesis [32,54]. Exosomes carry a wide expanse of miRNAs responsible for various processes, including signal transduction, self-antigen presentation on MHCIs, intercellular adhesion markers, and surface markers [54]. Other methods of interaction with receptor cells include receptor–ligand interaction; lipid-raft, caveolae, receptor, and clathrin-mediated endocytosis; micropinocytosis, and phagocytosis [103]. Manipulation of these sorting, signaling, and identification characteristics are potential opportunities for modification of exosomes function and reception by receptor cells [54].

Preconditioning can be performed by exposing MSCs to chemical agents, hormones, and physiological stresses. Specific treatments that skew MSC gene expression towards cardioprotective profiles include 2,4-dinitrophenol, prolyl hydroxylase, adiponectin, and hypoxic conditions, as well as cold storage [57,104,105,106,107]. Although preconditioning usually modifies MSCs to release the paracrine beneficial angiogenic and cytoprotective factors, accumulated evidence also indicate that preconditioning can impact the process of exosome synthesis to enrich specific miRNA in MSC-Exo. Preconditioning of platelet-derived growth factor (PDGF)-BB, a potent mitogen of MSCs, results in upregulation of miR-16-2 [108]. Of note, transplanted MSCs encounter a hostile microenvironment in the infarcted heart, which significantly stresses implanted MSCs and causes most cells to die. The poor survival of MSCs limits the therapeutic value of implanted MSCs. Ischemic preconditioning (IPC) enhances miR-107 in MSCs, leading to a significantly improved survival of transplanted MSCs in the infarcted myocardium [109]. IPC also effectively upregulates miR-210 expression and predominantly improves MSC survival following engraftment in a rat model of acute MI [110]. Markedly, upregulated miR-146a during preconditioning with diazoxide also plays a key role in promoting MSC survival [111].

In addition to improving stem cell survival, miRNAs regulated by preconditioning affect the function of MSC-Exo. Tanshinone IIA pretreatment has been reported to increase the content of miR-223-5p in MSC-Exo. Enriched miR-223-5p inhibits CCR2 activation to reduce monocyte infiltration in the I/R-injured heart, thus promoting cardioprotective effects in a rat MI/R model [96]. IPC of MSCs is shown to enhance miR-22 content, and significantly alleviate myocardial infarct size after MI [77]. Preconditioning to modify MSC exosomal miRNAs is simplified in Figure 2A.

### 4.2. Bioengineering

The content of exosomes has been observed to vary based on the type of secreting cell and on stresses that secreting cells have been subjected to. On the other hand, bioengineering strategies can be used to tailor the content of exosomes to include selective cargo (Figure 2B). Methods to manipulate exosome content can target the exosomes directly or the cell-secreting exosomes [112]. Direct methods of exosome loading include simple incubation with the target molecule, electroporation [44], and saponin permeabilization to increase membrane permeability and optimize target entry into exosome, and chemical conjugation of the target molecule (transfection) with agents such as lipofectamine to increase entry through the membrane [103]. Direct transfection of BM-MSC exosomes with miR19a/19b conferred increased cardiac recovery and decreased cardiac fibrosis compared to non-enriched BM-MSC exosomes in a mouse model [113]. These studies establish that loading exosomes directly can be an effective method of delivering specific cargo into exosomes. Additionally, concern for lack of exosome functionality following loading is unwarranted, as these enriched exosomes are still able to enact significant benefits in cardiac repair [18,113]. However, each of these methods is optimized for characteristics of loading molecules, such as lipophilicity or hydrophilicity, and may not be successful in loading more than one type of cargo into the same exosome.

Engineering reprogramming and transduction have been harnessed to alter expression in MSCs and composition of secreted exosomes [114,115]. Transduction of plasmid, lentivirus, and adenovirus vectors into host cells has also been a successful method to selectively alter exosome composition in MSCs [115,116]. Once in the cell, transcripts and proteins are at the mercy of cellular mechanisms for expression and inclusion into exosomes. However, although transduction does not allow for precise control of inclusion criteria for exosomes, it has generally been a successful method to induce overexpression of a transcript of interest, and expression of these molecules has been reflected in the composition and therapeutic activity of exosomes. Increasing GATA4 expression in MSCs [116] was able to enhance the levels of cardioprotective miRNAs present in exosomes derived from these cells [116]. Exosomes derived from MSCs overexpressing GATA4 were found to have greater expression of several miRNAs, including miR-19a and miR-451, compared to exosomes derived from control MSCs. A higher level of miR-19a in exosomes derived from GATA4 overexpressing MSCs, contributed to the cardioprotective effects of the exosomes. Similar augmented protective functions have also been observed from exosomes derived from MSCs genetically modified to overexpress HIF-1a and BMP2 [117,118]. Importantly, though the proteins overexpressed in the MSCs are not increased in secreted exosomes, the exosomes are found to have enriched miRNA content compared to exosomes derived from non-overexpressing MSCs [118]. In addition to altering exosome contents, bioengineering can alter exosome secretion. Increased exosome secretion is observed in MSCs overexpressing HIF-1a [107]. As expected, miRNA content is altered in these exosomes with increased secretion. Upregulated miRNAs include miR-15, miR-16, miR-17, miR-31, miR-126, miR-145, miR-221, miR-222, miR-320a, and miR-424. Together, the differential miRNA profiles in exosomes from HIF-1a overexpressing MSCs notably encourage Notch signaling, glycolysis/gluconeogenesis, and angiogenesis [107].

Bioengineering can be utilized to further target exosomes to recipient tissues of interest. In a murine model, manipulating transmembrane proteins on the exosome membrane allowed for successful targeting of EGFR-expressing breast cancer xenograft tissues [119]. Additionally, as exosome transmembrane proteins are a sampling of the membranes of the cells they were derived from, alteration of host cells to express certain markers is another method to tailor the identity of ligands and receptors on exosome membranes [120]. Vector delivery of cardiac targeting peptide (CTP)-Lamp2b into HEK 293 cells and glycosylation sequence stabilization increased expression of cardiac targeting peptide (CTP)-Lamp2b on exosome surfaces. Administration of exosomes expressing CTP-Lamp2b increased in vivo delivery of exosomes to mice hearts by 15% compared to controls lacking CTP (expressing Lamp2b only) [121]. These findings establish that bioengineering is a viable way to target exosome destinations and increase therapeutic doses reaching ischemic cardiac tissue to encourage pathways of cardiac repair.

## 5. Prospective and Limitation of MSC-Exo-Based Therapy in Cardiac Repair

Exosomes have numerous characteristics that optimize them for therapeutics. In [122], the following are cited as therapeutic benefits of exosomes: small size, low complexity, lack of nuclei and concurrent decreased risk of neoplastic transformation, stability, ease of production, long preservation, and potential for loading proteins, small molecules, or RNAs for delivery of biomolecules. Exosomes also contain significantly less DNA and MHC content compared to cell-based treatments, decreasing adverse immunogenic activation [19,120]. Additionally, due to their small size and lipophilic membrane, they can be absorbed and reach deeper tissues than other therapies. Exosomes even possess the ability to cross the blood brain barrier [123]. Furthermore, exosomes can overcome disadvantaged issues of using MSCs in cardiac repair, including the low cell survival, arrhythmia and tumorigenesis risk, etc.

The heart contains different types of cells, including cardiomyocytes, endothelial cells, and cardiac fibroblasts, etc. During injury, these cells can release exosomes to modulate the function of other cells in response to the stress. It has been shown that cardiomyocyte-secreted exosomal miR-222 and miR-143 promote cardiac angiogenesis during ischemia [124]. Cardiomyocyte exosomal miR-217 also contributes to increased cardiac fibroblast proliferation and cardiac fibrosis [125]. Upregulated miR-218-5p is observed in exosomes derived from the late-stage familial dilated cardiomyopathy patients and is a critical contributor to fibrogenesis through the suppression of TNFAIP3 activated TGF-β signaling [126]. In addition to cardiomyocyte-derived exosomes, miR-21-3p containing exosomes from cardiac fibroblasts facilitate development of cardiac hypertrophy when delivered into cardiomyocytes [127]. Myofibroblast-derived exosomes impart miR-200a-3p upon endothelial cells, impairing the angiogenic potential of endothelial cells via inhibition of placental growth factor production [128]. Furthermore, miR-214-containing exosomes from endothelial cells promote tube formation by inhibition of the cell cycle arrest in recipient endothelial cells [129]. Considering immune cell-derived exosomes, activated macrophage exosomal miR-155 in the infarcted myocardium inhibits fibroblast proliferation via reducing Son of Sevenless 1 but increases inflammation by downregulating Suppressor of Cytokine Signaling 1 [130]. M1 macrophage-secreted exosomal miR-155 has also been shown to impair the angiogenic ability of endothelial cells after MI [131]. Activated CD4+ T cells further worsen cardiac fibrosis through the exosomal miR-142-3p-WNT pathway [132]. Interestingly, although miRNA-containing exosomes play an important role in communications among different types of cells that respond to stress in the heart, no direct information is available regarding whether and how endogenous cardiac cell-derived exosomes interact with the effect of MSC-Exo after heart injury. Indeed, human cardiomyocyte-derived exosomes have been shown to induce human MSCs expressing cardiac genes in vitro [133]. This suggests the potential of tissue-specific exosomes in the induction of lineage-determined differentiation of MSCs and that cardiac cell-derived exosomes could impact the therapeutic value of MSCs. Nevertheless, further study on this topic will better guide the clinical applications of MSC-Exo for cardiac repair and regeneration.

It is noteworthy that an important limitation of exosomes is the absence of FDA-approved exosome-based products. To date, there is no standard approach in producing an efficient yet pure yield of exosomes. Though ultracentrifugation is the most frequently used method to isolate exosomes in study protocols, exosome RNA content, membrane damage, impurity inclusion, and throughput yield remains inconsistent and unoptimized between collections [134]. Current mainstream methods of purification include centrifugation followed by ultrafiltration or size exclusion chromatography. Exosome precipitation, sequential filtration, microfabrication technology, and immune-affinity capture are additional methods used for purification [135]. These methods manipulate size, density, immunoaffinity, acoustic, electrophoretic, electromagnetic properties of exosomes [135]. Combining purification methods, such as size exclusion chromatography with subsequent immune-affinity capture, has proven to be quite specific in isolating exosomes [136]. Yet, even these combination isolation methods that are more successful at isolation than any one single method alone still include some degree of non-exosome protein contamination. Common exosome sample contaminants include the following free proteins: albumin, immunoglobulins, and matrix metalloproteases [137,138]. Looking past contaminants, complications of long processing times (hours) and low purification product yield remain [139]. A study of exosome protein purification found that processing 1 cm of culture medium can be expected to yield less than 1 μg of exosomal protein. This contrasts with the 10–100 μg protein typically comprising a therapeutic dose [140]. Though combinatorial and targeted isolation methods have been quite successful in purifying samples, more research is required to increase pure exosomal yield to produce the required therapeutic levels. In fact, clinical applications of exosomes are limited due to the difficulty of generating a large-scale production of exosomes. In this regard, plant-derived EVs are much easier to scale up [141] and could be a promising alternative to deliver specific miRNAs of interest for cardiac repair and regeneration.

Yet other hurdles remain in the development of MSC-Exo-based therapeutics. Heterogeneity of MSCs results in a variable population of secreted exosomes, likely leading to variable therapeutic effects on cardiac repair and regeneration. Furthermore, clinical delivery of exosomal miRNAs for cardiac repair is hindered by off-target delivery and limited retention of therapeutic exosomal doses at the site of damaged myocardium [142]. Exosomes could reach not only the infarcted myocardium, but also other organs/tissue such as lungs, when administered systemically. Studies endorse the rapid redistribution of exosomes to other sites, such as the spleen, GI tract, and lungs, with redistribution rates and patterns dependent on delivery type and cell type in animal models [143]. As described previously, a bioengineering approach can be used to modify exosome membrane proteins for increased targeting of specific cells/tissue of interest [119,121]. However, there remains progress to be made in optimizing the delivery of therapeutic levels at damaged cardiac tissues. To address suboptimal dose delivery to cardiac tissue, cardiac patches and hydrogels have been found to maintain a slow releasing yet potent dose of exosomes to encourage endogenous repair. While exosome retention via intramyocardial injection delivery only lasted three hours [144], hydrogel application was found to maintain local exosome concentrations for weeks [145]. A recent study further indicates that combining biomaterials with MSC-Exo improves their targeting and retention in the injury site for cardiac repair as shown in a rat MI model [146]. Similar to liposomes used to deliver drugs, exosomes can be modified with polyethylene glycol, which increases time in circulation (decreases sequestration by non-targeted tissues) and reduces uptake in non-targeted tissues [147]. These methods subdue concerns of subtherapeutic levels at damaged tissues due to redistribution and side effects due to off-target effects.

## 6. Conclusions

Accumulated evidence suggests that the therapeutic value of MSCs is mainly attributable to exosomes. Despite the complexity of exosomal miRNAs, MSC-Exo has emerged as a promising candidate for cardiac repair and regeneration. MSC-Exo replicates the beneficial effects of MSCs by regulating various cellular responses and signaling pathways implicated in cardiac regeneration and repair. MSC-Exo contains the enrichment of cardiac repair-favorable miRNAs, being able to provide anti-apoptotic, anti-inflammatory, anti-fibrotic, and angiogenic effects within the injured heart. MSCs can be preconditioned and bioengineered to express or carry specific miRNAs of interest for better outcomes concerning cardiac repair. Studying exosomal miRNAs will provide important insight into the molecular mechanisms of MSC-Exo in cardiac regeneration and repair. This significant information can help optimize cell-free treatment and overcome the challenges associated with MSC-Exo therapeutic application.

Despite the evident benefits, MSC-Exo has not yet been clinically adapted to treat heart disease. To date, all clinical applications of MSC-Exo (seven published clinical trials and fourteen ongoing clinical studies) are related to respiratory disease (ARDS, pulmonary infection), immune system dysfunction (graft-versus-host disease, periodontitis, osteoarthritis, Covid-19), Alzheimer’s disease, stroke, and type I diabetes [148,149]. Using MSC-Exo to conduct cardiac repair in clinical practice requires optimizing the isolation methods, administration route, and dosing. Notably, current technical limitations lead to a difficulty in isolating and purifying exosomes from microvesicles and apoptotic bodies, which may complicate the functions of MSC exosomal miRNAs in cardiac repair. Additionally, the heterogeneous population of exosomes raises a concern about the exact mechanism underlying exosomal miRNA-based therapies. Furthermore, systemic administration of MSC-Exo miRNAs may cause off-target effects. Nonetheless, future investigation could focus on how to modify MSC-Exo to increase their target and retention in the injured site. More research is also needed to fully understand the mechanisms and potential benefits of the applications of MSC exosomal miRNAs in cardiac repair and regeneration.

## Figures and Tables

**Figure 1 cells-12-02815-f001:**
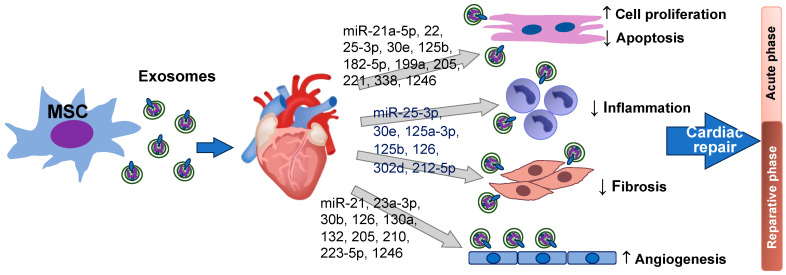
miRNAs derived from MSC exosomes improve cardiomyocyte survival/proliferation, promote angiogenesis, suppress inflammation, and reduce cardiac fibrosis in the heart after injury. Except the heart image from Heart Research Institute, this figure is created by the authors using PowerPoint.

**Figure 2 cells-12-02815-f002:**
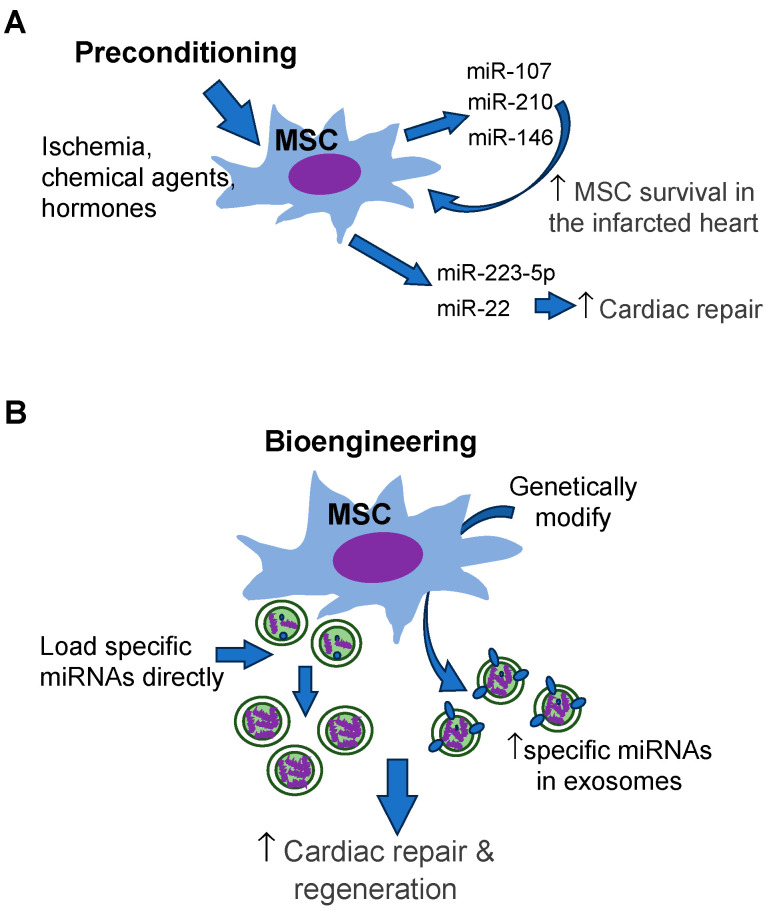
Modification of MSCs to enrich cardiac repair-favorable miRNAs so as to improve cardiac regeneration and repair. (**A**) Preconditioning of MSCs. (**B**) Bioengineering of MSCs. This figure is created by the authors using PowerPoint.

**Table 1 cells-12-02815-t001:** MSC exosomal miRNAs in cell proliferation and survival.

miRNA	Cell Source	Experimental Model	Mechanism	Year Published	Reference
miR-21a-5p	hBM-MSCs	hESC- & hiPSC-derived CMs, in vitro	Improving cardiac contractility & Ca2+ handling via PI3K	2018	[74]
	hMSCs	H9c2 cells, in vitro; Mouse MI/R, in vivo	↓ PTEN, PDCD4, Peli1, and FasL	2018	[75]
miR-22	Mouse BM-MSCs	Mouse MI, in vivo	Targeting methyl CpG binding protein 2 (Mecp2)	2014	[77]
miR-25-3p	Mouse BM-MSCs	Adult mouse CMs + OGD, in vitro; Mouse MI/R, in vivo	↓ FASL/PTEN	2020	[78]
miR-30e	Rat BM-MSCs	H9c2 cells + OGD, in vitro; Rat MI, in vivo	↓ LOX1/NF-κB p65/Caspase-9	2021	[79]
miR-125b	MSCs	Neonatal mouse CMs + hypoxia, in vitro; Mouse MI, in vivo	↓ p53-Bnip3	2018	[80]
miR-182-5p	Mouse BM-MSCs	Neonatal mouse CMs + sI/R, in vitro	↓ Gasdermin D	2022	[85]
miR-199a	hBM-MSCs	CMs from 18d mouse hearts + H_2_O_2_, in vitro	↓ Crim1, ↓ p53/ NF-κB, ↓ Bax	2018	[28]
miR-199a-3p	hBM-MSCs	Mouse MI/R, in vivo	---	2022	[35]
		Adult mouse CMs, mouse hearts, in vitro & in vivo	Cell cycle re-entry	2012	[66]
miR-205	Mouse Ad-MSCs	Neonatal rat CMs, in vitro; mouse MI, in vivo	↓ production of ROS	2023	[81]
miR-221	Rat BM-MSC	Neonatal rat CMs + hypoxia, in vitro	↓ p53 and PUMA	2013	[20]
miR-221-3p	hBM-MSCs	H9c2 cells + hypoxia, in vitro; Rat MI, in vivo	PTEN/Akt pathway	2020	[82]
miR-338	Rat BM-MSCs	H9c2 cells + H_2_O_2_, in vitro; Rat MI	↓ BAX/BCL2	2020	[83]
miR-1246	hUC-MSCs	H9c2 + hypoxia, in vitro; Rat MI	targeting serine protease 23 (PRSS23); ↓ the Snail/alpha-smooth muscle actin signaling	2021	[84]

h: human; BM: bone marrow; Ad: adipose tissue; UC: umbilical cord; CM: cardiomyocyte; MSC: mesenchymal stem cell; ESC: embryonic SC; iPSC: inducible pluripotent SC; MI: myocardial infarction/ischemia; MI/R: myocardial ischemia/reperfusion; sI/R: simulated I/R; OGD: oxygen-glucose deprivation.

**Table 2 cells-12-02815-t002:** MSC exosomal miRNAs in angiogenesis.

miRNA	Cell Source	Model	Mechanism	Year Published	Reference
miR-21	hEnMSCs, BM-MSCs, Ad-MSCs	Rat MI	↑ PTEN/Akt pathway	2017	[76]
miR-23a-3p	hUC-MSCs	HUVECs	↑ PTEN/Akt pathway	2022	[21]
miR-30b	MSC cell line	HUVECs	↓ DLL4	2017	[95]
miR-126	Rat Ad-MSCs	EPCs (Matrigel assay); Rat MI	Possibly ↓ inflammation	2017	[92]
miR-130a-3p	hBM-MSCs	HUVECS	↓ anti-angiogenic homeobox gene *HOXA5*	2018	[28]
miR-132	Mouse BM-MSCs	HUVECS; Mouse MI	↓ RASA1	2018	[94]
miR-205	Mouse AD-MSC	Neonatal rat CMs; mouse MI	↑ HIF-1α and VEGF	2023	[82]
miR-210	Mouse BM-MSC	HUEVCs, Mouse MI	↓ Efna3	2017	[93]
miR-223-5p	hUC-MSCs	HUEVCs Rat MI/R	↓ CCR2 activation and monocyte infiltration	2023	[96]
miR-1246	hUC-MSC	HUEVCs Rat MI	targeting serine protease 23 (PRSS23); ↓ the Snail/alpha-smooth muscle actin signaling	2021	[84]

h: human; En: endometrium; BM: bone marrow; Ad: adipose tissue; UC: umbilical cord; CM: cardiomyocyte; HUVECs: human umbilical vein endothelial cells; EPC: endothelial progenitor cell cardiomyocyte; MI: myocardial infarction/ischemia.

## Data Availability

Not applicable.

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
