# Peer review of "Mesenchymal Stem Cell-Derived Exosomal microRNAs in Cardiac Regeneration"

_cells, 2023, doi:10.3390/cells12242815_

Round 1
Reviewer 1 Report
Comments and Suggestions for Authors
This article by Bhaskara et al, seems comprehensive in exploring the role of MSC-derived exosomal miRNAs benefits in cardiac repair and regeneration. The article is well-structured and provides a comprehensive overview. I only have minor comments, in no order of magnitude.
· Reference is missing where the MSC-Exo prevents excessive scar formation after MI, at line 312.
· Another important study to be included, Controlled Inhibition of the Mesenchymal Stromal Cell Pro-inflammatory Secretome via Microparticle Engineering (PMID- 27264972)
· Can authors provide or add another minor section/table describing the completed or ongoing specific MSCs exo/microRNA-based clinical trials for cardiac disease?
· Minor corrections-
At line 123, “miRNA signaling has a powerful,
At line 124, human miRNA can alter the expression
At line 408,.. observed to vary based on the type of secreting cell
At line 473, A higher level of
Author Response
We sincerely thank you for reviewing our manuscript and for providing insightful suggestions. We have made all revisions along the lines suggested and addressed each below.
- Reference is missing where the MSC-Exo prevents excessive scar formation after MI, at line 312.
Response: We apologize for our omission on this. References have been added to the article (line 346).
- Another important study to be included, Controlled Inhibition of the Mesenchymal Stromal Cell Pro-inflammatory Secretome via Microparticle Engineering (PMID- 27264972)
Response: Thank you for the suggestion. We have discussed this important study (line 351 – 353).
- Can authors provide or add another minor section/table describing the completed or ongoing specific MSCs exo/microRNA-based clinical trials for cardiac disease?
Response: Thank you for the suggestion. We have added several sentences to discuss it (line 600 – 606).
- Minor corrections-
At line 123, “miRNA signaling has a powerful,
At line 124, human miRNA can alter the expression
At line 408,.. observed to vary based on the type of secreting cell
At line 473, A higher level of
Response: Thank you for pointing these out. We have corrected them (line 123, 124, 439, and 468).
Reviewer 2 Report
Comments and Suggestions for Authors
Overall, the manuscript titled "Characteristics and Potential Mechanisms of MSC-Derived Exosomal miRNAs in Cardiac Repair and Regeneration" provides a comprehensive overview of the therapeutic potential of mesenchymal stem cell (MSC)-derived exosomes in cardiac repair. The authors highlight the importance of exosomes as mediators of the beneficial effects of MSCs and emphasize the role of specific microRNAs (miRNAs) carried by MSC-derived exosomes in promoting cardiac regeneration and repair.
The manuscript is well-written and organized, and the content is highly relevant to the field of regenerative medicine. The authors effectively summarize the existing knowledge on this topic and present a cohesive narrative. However, it is important for the authors to highlight the novel aspects of this study, especially considering that the topic of MSC-derived exosomes and their role in cardiac repair has been previously covered in the literature.
To enhance the originality and novelty of the manuscript, I suggest the authors emphasize the following points:
1. Novel miRNAs: Discuss any recently discovered or less-explored miRNAs that have been identified in MSC-derived exosomes and their potential roles in cardiac regeneration. Highlighting the discovery of unique or less-studied miRNAs will contribute to the existing knowledge base and attract readers' attention.
2. Optimization of Cell-Free Treatment: Discuss the current challenges associated with the therapeutic application of MSC-derived exosomes and propose innovative strategies to overcome these hurdles. Highlight any recent advancements or cutting-edge techniques that have been developed to enhance the efficacy and delivery of exosomal miRNAs for cardiac repair.
he accurate characterization and classification of extracellular vesicles (EVs), including exosomes, is indeed a challenge in the field. The MISEV2018 guidelines provide valuable recommendations for the proper classification and characterization of EVs, including the use of appropriate terminology.
To ensure clarity and accuracy in the manuscript, I recommend that the authors adhere to the MISEV2018 guidelines when referring to exosomes and other types of EVs. It is crucial to accurately describe the isolation methods, size, markers, and functional characteristics of the vesicles under investigation to avoid confusion and misinterpretation.
exosomes contain a mixture of components, and it is possible that some of these components may have negative effects on other areas or tissues in the body. However, it should be noted that many current studies indicate that exosomes produced by mesenchymal stem cells (MSCs) exhibit diverse and positive characteristics in various therapeutic contexts.
In order to ensure the safety and efficacy of exosome-based therapies, the effects of exosomes on other bodies and tissues should be studied and documented. There are studies that shed light on the safety of exosomes and evaluate them in animal models and clinical trials. Through such studies, components with negative effects can be identified, modified, or excluded from the exosomes used in therapy.
Therefore, it is recommended that researchers pay great attention to studying the potential negative effects of exosomes on other tissues, documenting their safety and efficacy. Such studies can help improve the selection and preparation of stable and safe exosomes for therapeutic use, ensuring the avoidance of any undesirable negative effects on other tissues in the body.
The study by Alzahrani et al. titled "Plant-Derived Extracellular Vesicles and Their Exciting Potential as the Future of Next-Generation Drug Delivery" highlights the emerging field of plant-derived extracellular vesicles (EVs) as a promising new generation of therapeutics. The authors provide valuable insights into the potential use of exosomes isolated from plant sources for drug delivery purposes.
Considering the relevance of this study to both the general field of exosome-based therapeutics and the specific topic of cardiac repair, I highly recommend that the authors include a discussion or reference to the study by Alzahrani et al. (2023) in their manuscript. By doing so, the authors can enhance the comprehensiveness of their review and provide readers with a broader understanding of the expanding field of exosome-based therapeutics, particularly in the context of cardiac repair.
An important limitation of exosomes is the absence of FDA-approved exosome-based products, unlike liposomes that have received approvals for certain therapeutic applications. To provide a comprehensive analysis, I highly recommend that the authors explicitly acknowledge this limitation in either the limitations section or the discussion of their review. By addressing the lack of FDA-approved exosome products, the authors can effectively address the current status of exosome-based therapeutics and shed light on the regulatory challenges they encounter.
Moreover, it would be beneficial for the authors to discuss the potential reasons behind the absence of FDA approvals for exosome products. Factors such as the complexity of exosome isolation, purification, and characterization could be highlighted, along with the need for additional clinical trials and the establishment of regulatory guidelines to ensure safety and efficacy.
By addressing these considerations, the authors can provide a well-rounded evaluation of the limitations and regulatory hurdles associated with exosome-based therapeutics. Such a discussion will contribute to a more comprehensive understanding of the current state of exosome research and development, while also emphasizing the need for further studies and regulatory advancements in this field.
Author Response
We sincerely thank you for reviewing our manuscript and for providing insightful suggestions. We have made all revisions along the lines suggested and addressed each below.
- Novel miRNAs: Discuss any recently discovered or less-explored miRNAs that have been identified in MSC-derived exosomes and their potential roles in cardiac regeneration.
Response: Thank you for the suggestion. We have discussed recently discovered or less-explored MSC exosomal miRNAs and their potential roles in cardiac repair, including miR-25-3p, miR-30e, miR-182d-5p, miR-302d-3p, and miR-212-5p (Table 1; line 230-236, 248-250, 254-256, and 373-383). These studies are from 2020 and later.
- Optimization of Cell-Free Treatment: Discuss the current challenges associated with the therapeutic application of MSC-derived exosomes and propose innovative strategies to overcome these hurdles.
Response: Thank you for the suggestion. We have comprehensively discussed the current challenges/limitations associated with the therapeutic value of MSC-derived exosomes and proposed potential strategies to overcome these hurdles (line 537 – 586). We have also included the study by Alzahrani et al. in our discussion (line 559 – 562).
Reviewer 3 Report
Comments and Suggestions for Authors
Suggestions:
1. Adding a couple of figures/diagrams to visualize cardiomyocyte proliferation, angiogenesis, and fibrosis could help the audience better understand the roles of exosome in cardiac repairt.
2. In the final discussion part, please comment on roles of exosome derived from different type of cardiac cells after heart injury. These exosomes could limit the effect of MSC-derived exosome.
3. When referring to a previous report, please describe the animal model and injury type used in that report.
4. Cardiomyocyte proliferation is a vague term. In those reports, are proliferating cardiomyocytes marked by pHH3, Ki67, or AuroraKb? please consider these when describe cardiomyocyte proliferation since cardiomyocytes may re-enter cell cycle, but most of them do not finish the cytokinesis.
5. Please comment on and compare the effect of different miRNAs on cardiomyocyte cell cycle activity, angiogenesis etc. Adding a bar graph would be helpful.
Comments on the Quality of English LanguageThe English writting is very good. However, there are duplicated information about a certain miRNA that may impact the flow of the maintext.
Author Response
We sincerely thank you for reviewing our manuscript and for providing insightful suggestions. We have made all revisions along the lines suggested and addressed each below.
- Adding a couple of figures/diagrams to visualize cardiomyocyte proliferation, angiogenesis, and fibrosis could help the audience better understand the roles of exosome in cardiac repair.
Response: Thank you for the suggestion. We have added one comprehensive figure (Figure 1) to better deliver the roles of MSC exosomal miRNAs in cardiomyocyte proliferation, angiogenesis, and fibrosis for cardiac repair.
- In the final discussion part, please comment on roles of exosome derived from different type of cardiac cells after heart injury. These exosomes could limit the effect of MSC-derived exosome.
Response: We have added one paragraph to discuss roles of cardiac cell-derived exosomes in intercellular communication after heart injury (line 507 – 536).
- When referring to a previous report, please describe the animal model and injury type used in that report.
Response: We apologize for this omission. We have added such information.
- Cardiomyocyte proliferation is a vague term. In those reports, are proliferating cardiomyocytes marked by pHH3, Ki67, or AuroraKb? please consider these when describe cardiomyocyte proliferation since cardiomyocytes may re-enter cell cycle, but most of them do not finish the cytokinesis.
Response: Thank you for pointing it out. We have added more details to discuss cardiomyocyte proliferation and re-entry of cell cycle.
- Please comment on and compare the effect of different miRNAs on cardiomyocyte cell cycle activity, angiogenesis etc. Adding a bar graph would be helpful.
Response: Thank you for the suggestion. This information is added in a new Figure 1.
--- The English writing is very good. However, there are duplicated information about a certain miRNA that may impact the flow of the main text.
Response: Because the same exosomal miRNA could regulate different processes (more than one action) for cardiac repair, we do discuss a certain miRNA in different sections. We have added some sentences to make the flow of the main text more logically (line 317 – 318 and 367 – 369).